# Erythrocyte Membrane Cloaked Curcumin-Loaded Nanoparticles for Enhanced Chemotherapy

**DOI:** 10.3390/pharmaceutics11090429

**Published:** 2019-08-23

**Authors:** Xiaotian Xie, Haijun Wang, Gareth R. Williams, Yanbo Yang, Yongli Zheng, Junzi Wu, Li-Min Zhu

**Affiliations:** 1College of Chemistry, Chemical Engineering and Biotechnology, Donghua University, Shanghai 201620, China; 2UCL School of Pharmacy, University College London, 29-39 Brunswick Square, London WC1N 1AX, UK; 3College of Basic Medicine, Yunnan University of Traditional of Chinese Medicine, Kunming 650500, China

**Keywords:** erythrocyte membrane, porous PLGA nanoparticle, curcumin, anticancer therapeutics

## Abstract

In this study, curcumin-loaded porous poly(lactic-*co*-glycolic acid) (PLGA) nanoparticles (NPs) were prepared and surface modified with red blood cell membranes (RBCM) to yield biomimetic RBCM-p-PLGA@Cur NPs. The NPs displayed a visible cell-membrane structure at their exterior and had a uniform size of 162 ± 3 nm. In vitro studies showed that drug release from non-porous PLGA NPs was slow and that much of the drug remained trapped in the NPs. In contrast, release was accelerated from the porous PLGA NPs, and after the RBCM coating, a sustained release over 48 h was obtained. Confocal microscopy and flow cytometry results revealed that the RBCM-p-PLGA NPs led to a greater cellular uptake by H22 hepatocarcinoma cells than the uncoated analogue NPs, but could avoid phagocytosis by macrophages. The drug-free formulations were highly biocompatible, while the drug-loaded systems were effective in killing cancer cells. RBCM-p-PLGA@Cur NPs possess potent anti-tumor activity in a murine H22 xenograft cancer model (in terms of reduced tumor volume and mass, as well as inducing apoptosis of tumor cells), and have no observable systemic toxicity. Overall, our study demonstrates that the use of the RBCM to cloak nanoscale drug delivery systems holds great promise for targeted cancer treatment, and can ameliorate the severe side effects currently associated with chemotherapy.

## 1. Introduction

Despite many recent therapeutic innovations, it remains the case that cancer imposes a major disease burden worldwide. Traditional methods of treatment include surgery, chemotherapy, and radiation therapy. However, it is not possible to surgically resect all tumors, and chemotherapy is associated with severe off-target effects. Major research efforts are underway to develop effective and non-toxic drug carriers able to target therapy specifically to tumor tissue, and thus to ameliorate these challenges in the treatment of cancer [1,2]. 

There exist a number of approaches that can be used to achieve such targeting. The simplest involves passive targeting, relying on the build-up of a drug delivery system (DDS) in the tumor because of its leaky vasculature [3]. Active targeting can also be realized; this has mostly been achieved by functionalizing particles with ligands to membrane receptors over-expressed on cancer cells, thus aiding specific uptake of a formulation by the target cell population [4,5,6,7]. Biomimetic systems have also been developed, with the aim of improving circulation times, avoiding an immune response, and/or increasing targeting specificity [8]. The latter approach involves attempting to disguise DDSs as endogenous cells or molecules [9]. 

Red blood cells (RBCs, otherwise known as erythrocytes) comprise the most abundant population of cells in the blood, and have attracted increasing attention in medicines development [10,11]. RBCs have extended systemic circulation times [12], and being endogenous cells are highly biocompatible with low immunogenicity. They can also be used in the development of DDSs, and using the RBC membrane (RBCM) to cloak nanoparticle DDSs has been deemed an effective approach to developing improved therapeutics [13,14]. The presence of the protein CD47 on the RBCM regulates macrophage uptake by interacting with the CD47 receptor-signaling protein on the surface of the latter [15]. Consequently, RBCM-coated materials can act as effective drug carriers in vivo, avoiding phagocytosis and persisting in the systemic circulation for a long period of time. 

There are myriad of compounds reported to have anti-cancer activity. One which has been especially widely explored is curcumin (Cur) [16], a phenolic pigment extracted from the rhizome of *Curcuma longa*. Cur has been found to have a wide range of pharmacological activities, including anti-inflammatory and anti-oxidant effects [17]. In addition, it inhibits tumor growth [18,19], lowers blood cholesterol levels [20], promotes wound healing [21], and also has immunomodulation [22], antibacterial [23] and anti-fibrotic effects [24]. However, there are a number of obstacles that need to be overcome before it can be widely used in therapeutic applications: Cur has low stability, poor water solubility, low bioavailability, and undergoes rapid in vivo metabolism. As a result, novel drug delivery systems able to circumvent these issues are needed. 

The use of nanoparticulate carriers is one approach that has long been used to improve the pharmacokinetics of anticancer drugs, including Cur [25]. These can be fabricated from a wide range of materials, with the polymer poly(lactic-*co*-glycolic acid) (PLGA) having been particularly extensively studied [26]. PLGA is approved for use in humans by the United States Food and Drug Administration, since it has low toxicity and tunable biodegradability. However, drug release from monolithic PLGA nanoparticles is often slow, lasting weeks or months [27]. For anti-cancer applications, a more rapid release profile is likely to be required. To solve this problem, porous PLGA nanoparticles can be generated to accelerate drug release [28]. Vitamin E TPGS (TPGS) is one species which can be used to induce pore formation, and it has been found that the drug release profile can be tuned by adjusting the component ratio between PLGA and TPGS [29,30,31].

In this work, we developed a drug delivery system using the RBCM to cloak porous PLGA nanoparticles loaded with Cur (RBCM-p-PLGA@Cur NPs), and explored their effect on H22 cancer cells both in vitro and in vivo in an xenograft mouse model (Scheme 1). 

## 2. Materials and Methods

### 2.1. Materials and Reagents 

PLGA (lactide:glycolide ratio = 50:50, *M*_w_ = ~21,000) was purchased from Jinan Daigang Biological Engineering Co., Ltd. (Jinan, China). Cur and vitamin E TPGS (TPGS) were sourced from Aladdin (Shanghai, China). Hoechst 33342 dye, ethylenediamine, triethylamine, *N*-hydroxysuccinimide (NHS), 1-(3-dimethylaminopropyl)-3-ethylcarbodiimide hydrochloride (EDC·HCl), dimethylsulfoxide (DMSO), 4-dimethylaminopyridine (DMAP), fluoresceinisothiocyanate (FITC) and 3-(4,5-dimethylthiazol-2-yl)-2,5-diphenyl-tetrazolium bromide (MTT) were purchased from Sigma-Aldrich (St. Louis, MO, USA). A membrane protein extraction kit and protease inhibitor were procured from Phygene Life Sciences (Fuzhou, China), and a Pierce BCA protein assay kit from Life Technologies (Hudson, NH, USA). Trypsin, fetal bovine serum (FBS), RPMI-1640 medium, penicillin, streptomycin and phosphate buffered saline (PBS, pH = 7.4) were obtained from Gibco (Carlsbad, CA, USA). H22 cells (a murine hepatocellular carcinoma cell line), 4T1 cells (a mouse breast adenocarcinoma cell line), L929 cells (a mouse fibroblast cell line) and RAW264.7 (macrophage) cells were provided by the Institute of Biochemistry and Cell Biology of the Chinese Academy of Sciences (Shanghai, China). All chemicals and reagents were used as received. Deionized water (H_2_O) was purified with a Millipore system (Milli-Q, 18.2 MΩ cm, Burlington, MA, USA).

### 2.2. Preparation and Characterization of p-PLGA NPs 

Blank porous PLGA NPs (p-PLGA NPs) were prepared using the nanoprecipitation method, as described previously [31]. A total of 10 mg PLGA and 2 mg TPGS were dissolved in 1 mL acetone. The solution was sonicated until complete dissolution was achieved, and then added dropwise into 5 mL of deionized water under stirring at 1000 rpm. After leaving the suspension to stand for 3 h to evaporate the acetone [32], 500 μL of the resulting NP suspension was set aside for electron microscopy, dynamic light scattering (DLS) and zeta potential analysis, and the rest was stored at 4 °C for further use.

### 2.3. Loading of Cur 

For the preparation of p-PLGA@Cur NPs, 100 mg PLGA, 20 mg TPGS and 10 mg of Cur were dissolved in 10 mL acetone. The solution was sonicated until complete dissolution was achieved, and then added dropwise into 50 mL of deionized water under stirring at 1000 rpm. After 3 h, the obtained suspension was centrifuged at 12,000 rpm for 15 min to remove unloaded Cur. The supernatant was 10-fold diluted with methanol and a UV-vis spectrophotometer (UNICO, Shanghai, China) used to quantify the Cur loading. The Cur concentration was calculated based on a pre-determined calibration curve. The Cur encapsulation efficiency (EE %) and loading content (LC %) were determined as follows:Mass of Cur in NPs = total mass of Cur in feed − mass of Cur in the supernatant
Total mass of NPs = mass of polymer + mass of Cur in NPs
$$\mathrm{EE}\%\;=\;\frac{\mathrm{mass}\;\mathrm{of}\;\mathrm{Cur}\;\mathrm{in}\;\mathrm{NPs}}{\mathrm{total}\;\mathrm{mass}\;\mathrm{of}\;\mathrm{Cur}\;\mathrm{in}\;\mathrm{feed}}\times 100\%$$
$$\;\mathrm{LC}\%\;=\frac{\mathrm{mass}\;\mathrm{of}\;\mathrm{Cur}\;\mathrm{in}\;\mathrm{NPs}}{\mathrm{total}\;\mathrm{mass}\;\mathrm{of}\;\mathrm{NPs}}\times 100\%$$

### 2.4. Isolation of RBCM 

Cells were extracted from fresh whole mouse blood, and then transferred into an anticoagulant-coated tube [14,33,34]. First, the blood was centrifuged at 3000 rpm for 5 min at 4 °C to remove the plasma and leukocytes, then washed with phosphate buffered saline (PBS, pH = 7.4) three times. The RBCs obtained were resuspended in a 0.2 mM solution of EDTA in water, to induce membrane rupture. The hemoglobin released was removed by centrifugation at 14,800 rpm for 7 min at 4 °C. The resultant pellet was resuspended in 0.2 mM EDTA and centrifuged again. The EDTA/centrifugation steps were repeated until the supernatant was clear and colorless. Finally, the RBCMs were suspended in PBS (pH = 7.4) containing 0.1 mg/mL of a protease inhibitor and stored at −80 °C for further use. 

### 2.5. Preparation and Characterization of RBCM-p-PLGA@Cur NPs 

RBCM-p-PLGA NPs were prepared following a literature protocol [35,36]. Briefly, 2 mL of a 2 mg/mL suspension of p-PLGA NPs was mixed with RBCM derived from 500 μL of whole blood under magnetic stirring for 10 min. This mixture was then ultrasonicated at 100 W for 5 min. Finally, RBCM-p-PLGA NPs were obtained after the suspension was extruded repeatedly through 200 nm polycarbonate membranes. To prepare RBCM-p-PLGA@Cur NPs, the same procedure was performed using p-PLGA@Cur NPs as the starting material. The hydrodynamic particle size and zeta potential of the RBCM coated nanoparticles were measured using a Zetasizer (Malvern Instruments, Westborough, MA, USA) and their morphology was characterized by TEM. The colloidal stability of the nanoparticles was verified using a Zetasizer. Experiments were performed at 25 °C in different media, including Milli-Q water and PBS. 

To further verify successful coating of RBCM on p-PLGA NPs, the membrane proteins in RBCM and RBCM-p-PLGA NPs were analyzed by sodium dodecyl sulfate-polyacrylamide gel electrophoresis (SDS-PAGE). Samples were lysed in a SDS buffer and then processed with a BCA assay kit to determine the total protein concentrations. The lysates were then run on a 4–12% Bis-Tris 10-well minigel in a running buffer, using a BIO-RAD electrophoresis system (80 V, 2 h, BIO-RAD, CA, USA). The resulting polyacrylamide gel was stained with Coomassie brilliant blue for 1 h to permit visualization of the protein bands.

### 2.6. Characterization Techniques 

Zeta potential and dynamic light scattering (DLS) were quantified at 25 °C with the aid of a Zetasizer Nano ZS system (Malvern Instruments, Westborough, MA, USA). Transmission electron microscopy (TEM) images were recorded using a JEOL 2010F transmission electron microscope (Hitachi, Tokyo, Japan), while field emission scanning electron microscopy (FESEM) images were captured on a JEOL S-4800 instrument (Hitachi, Tokyo, Japan). UV-vis spectra were recorded at 25 °C on a UV-1800 spectrophotometer (UNICO, Shanghai, China). Confocal laser scanning microscopy (CLSM) images were collected on a LSM 700 microscope (Carl Zeiss, Jean, Germany). Flow cytometry analysis was performed on a flow cytometer (Becton Dickinson, CA, USA).

### 2.7. Drug Release 

For control purposes, non-porous PLGA NPs loaded with Cur (PLGA@Cur NPs) were generated. The preparation method was the same as described for p-PLGA@Cur, except that TPGS was not added. The release of Cur from the NPs was then evaluated using a dialysis method. A total of 2.0 mL of 2 mg/mL suspensions of PLGA@Cur, p-PLGA@Cur or RBCM-p-PLGA@Cur NPs were loaded into a dialysis bag (MWCO = 7,000 Da), and immersed in PBS (18.0 mL, pH 7.4). The samples were placed in a incubator at 37 °C with shaking (100 rpm). Light was excluded throughout the experimental period of 48 h. Periodically, 1 mL aliquots were taken from the external medium and replaced with an equal volume of fresh pre-heated buffer. The concentration of Cur released was measured by UV-vis spectroscopy at λ_max_ of 425 nm. Data are reported as mean ± standard deviation (S.D.), *n* = 4.

### 2.8. Cytotoxicity Measurements 

An in vitro cytotoxicity investigation was performed to evaluate the biocompatibility of blank p-PLGA and RBCM-p-PLGA NPs with L929 cells and H22 cells [37]. Cells were cultured in RPMI-1640 medium supplemented with penicillin (100 μg/mL), streptomycin (100 μg /mL), and 10% *v*/*v* heat-inactivated fetal bovine serum (FBS). 200 μL of cell suspension was seeded in each well of a 96-well plate (1 × 10^4^ cells/well) and incubated at 37 °C under a 5% CO_2_ atmosphere for 24 h. 

For biocompatibility experiments, the medium was removed from the cells and 200 µL of fresh medium containing different concentration of p-PLGA or RBCM-p-PLGA NPs (5, 10, 20, 50, 80, 100, 200, 500 μg/mL) was added to each well, before the cells were cultured for another 24 h. For cytotoxicity experiments, after removal of the initial medium, 200 µL of fresh medium containing different concentrations of free Cur (2.5, 5, 10, 20, 30, 40, 50, 100 μg/mL), or p-PLGA@Cur or RBCM-p-PLGA@Cur NPs (giving equivalent drug concentrations) was added to each well, before the cells were cultured for a further 24 h. After that, the medium was aspirated and 20 μL of MTT solution (5 mg/mL) and 180 μL of fresh medium was added to each well. The plate was incubated for an additional 3 h in darkness. The medium was then removed, 200 μL of DMSO was added to each well, and the plate was shaken at 100 rpm for 15 min. The absorbance of the wells was finally measured at 570 nm, using a microplate reader (Multiskan FC, Thermo Scientific, Hudson, NY, USA). The relative cell viability was calculated relative to an untreated cells control. Data are reported as mean ± S.D. Three independent experiments each containing three replicate wells per condition were performed.

### 2.9. Cellular Uptake 

In order to track the distribution of the RBCM-p-PLGA NPs visually, FITC was employed to label the PLGA NPs. To prepare labeled NPs, a total of 300 mg of PLGA, 1.8 mg of NHS, 2.9 mg of EDC and 1 mg of DMAP were dissolved in 10 mL of dichloromethane and stirred for 24 h at 25 °C to convert PLGA to PLGA-NHS. Afterward, 1.8 mg of ethylenediamine and 8.6 μL of triethylamine were added to the solution, which was stirred for another 3 h. Subsequently, 35 mg of FITC was added into the solution and stirring was performed for a further 12 h at 25 °C. A rotary evaporator was used to remove the dichloromethane, and 5 mL of *N*,*N*-dimethylformamide (DMF) added to dissolve the conjugate. The resultant solution was placed into a dialysis bag (MWCO = 7000 Da) and dialyzed against water for 3 days to remove unreacted FITC. Finally, the product was freeze-dried for 48 h to obtain FITC-PLGA. The resultant FITC-PLGA was then used to prepare labeled p-PLGA NPs (FITC-p-PLGA NPs) and RBCM coated FITC labeled PLGA NPs (RBCM-FITC-p-PLGA NPs) using the same procedure as described above.

The uptake of FITC-p-PLGA NPs and RBCM-FITC-p-PLGA NPs was observed using CLSM [38]. A total of 200 μL of a RAW 264.7, H22 cell or 4T1 cell suspension (5 × 10^4^ cells/well) and 800 μL of complete medium were added to each well of a 24-well cell culture plate and incubated at 37 °C under a 5% CO_2_ atmosphere for 24 h. After aspiration of the medium, the cells were treated with 900 μL of fresh medium and 100 μL of PBS or 2 mg/mL FITC-p-PLGA NPs or RBCM-FITC-p-PLGA NP suspensions. Following incubation for another 2 h, the cells were washed twice with PBS, and 1 mL of glutaraldehyde (2.5%) was added and the cells kept for 15 min at 4 °C. This was followed by counterstaining with 0.5 mL Hoechst 33342 (10 μg/mL) for 15 min at 37 °C. Finally, images were obtained by CLSM.

For flow cytometry analysis, 2.4 mL of a H22 cell or RAW 264.7 suspension (containing 1 × 10^5^ cells) was placed in each well of a 6-well cell culture plate and incubated at 37 °C under a 5% CO_2_ atmosphere for 24 h. After removal of the medium, 2 mL of fresh RPMI and 500 μL of PBS or suspensions of FITC-p-PLGA or RBCM-FITC-p-PLGA NPs (200 μg/mL) were added, and the cells cultured for 4 h. The cell pellet was collected by centrifugation at 1100 rpm for 3 min, the supernatant discarded, and the pellet dispersed in 500 μL of PBS. Cellular uptake was determined by flow cytometry.

### 2.10. In Vivo Murine Tumor Model

All animal experiments were undertaken with full authorization from the Committee for Experimental Animal Welfare and Ethics of Yunnan University of Traditional Chinese Medicine. (project identification code: R-062019031, approval date: 11 March 2019). A total of 20 ICR female mice (specific pathogen-free grade, 18–20 g) were acquired from Jiangsu KeyGEN BioTECH Co. Ltd., Jiangsu, China. Tumors were implanted by subcutaneous injection into the right front limb armpit of each mouse of 1 × 10^6^ H22 cells in 100 μL of PBS (pH 7.2–7.4).

### 2.11. In Vivo Antitumor Efficacy 

After H22 cell implantation, the tumors were allowed to grow until their volume (calculated as 0.5 × length × diameter^2^) reached ca. 120 mm^3^. The mice were then randomly divided into four groups (5 mice per group). Each mouse was intravenously injected, via the tail vein, with PBS, free Cur (dissolved in 1% DMSO), p-PLGA@Cur NPs or RBCM-p-PLGA@Cur NPs (at a Cur dose of 2 mg/kg) every two days for 16 days. The tumor volumes and body weight were recorded every two days. 

After 16 days, all the mice were sacrificed by cervical vertebra dislocation. The heart, liver, spleen, lung, kidney and tumor of the mice were immediately excised and washed with physiological saline. The tissues were fixed with a 4% aqueous paraformaldehyde solution and embedded in paraffin for sectioning. The sliced organ tissues (thickness: 4 mm) were mounted on glass slides, stained by hematoxylin and eosin (H&E) [39,40] and imaged with a Nikon DS-U3 digital microscope (Nikon, Tokyo, Japan). Tumor tissues were subjected to terminal deoxynucleotidyl transferase (TdT) mediated dUTP-digoxigenin nick end labeling (TUNEL) staining according to the manufacturer’s instructions, and images were obtained using a Pannoramic 250 digital scanner (PerkinElmer, Boston, MA, USA).

### 2.12. Statistical Analysis 

Differences between two groups were analyzed by Student’s *t* test and mean values were compared via one-way ANOVA; the significance level was defined as *p* < 0.05, with * denoting *p* < 0.05, ** *p* < 0.005 and *** *p* < 0.001.

## 3. Results and Discussion

### 3.1. Characterization of NPs 

The p-PLGA nanoparticles were found to be spherical in shape by TEM (Figure 1a). Their size was 139 ± 11 nm (Figure 1b). However, any porous structure was unclear due to the small size of the particles and limitations in the resolution of the microscope. To check for porosity, larger nanoparticles were thus prepared using the same methodology but with a slower stirring speed of 300 rpm (see Appendix A). Under these conditions the porous structure was clearly seen, and thus it can be concluded that the use of TPGS permitted porous particles to be produced. After the RBCM coating, a membrane structure was visible around the p-PLGA NPs, with an outer lipid bilayer shell ≈ 16 nm in thickness (Figure 1c). The particles increased in size to 169 ± 8 nm (Figure 1d). 

DLS revealed that the hydrodynamic size of the RBCM-p-PLGA NPs was 162 ± 3 nm (Figure 1e) and that they had a low polydispersity index (PDI = 0.102). They were somewhat larger and more polydisperse than the bare p-PLGA NPs (138 ± 4 nm, PDI = 0.047). As shown in Figure 1f, the RBCM-p-PLGA@Cur NPs exhibited excellent colloidal stability, and could be dispersed in both water and PBS without aggregation over time. The zeta potential decreased from −28.4 ± 1.4 mV with p-PLGA to −34.3 ± 1.2 mV after RBCM coating (Figure 1g), similar to the potential of the RBCM precursor. A more negative zeta potential arose with the RBCM owing to the presence of anionic phospholipid head groups at the two exterior sides of the membrane. One side of this interacted favorably with the PLGA, while the other side formed hydrogen bonds with the water continuous phase.

To verify the presence of an intact RBCM coating, we subjected the RBCM-p-PLGA NPs, to SDS-PAGE gel electrophoresis (Figure 1h). The RBCM-p-PLGA NPs displayed almost identical gels to the RBCM, confirming that they contained the same protein components and that the RBCM was preserved in the NPs. Taken together, all these data allow us to conclude that the p-PLGA NPs were successfully coated with a RBCM layer. The observations noted here are in line with those in the literature [13,14].

### 3.2. Drug Loading and Release Behavior 

UV-vis absorption spectra of p-PLGA and the p-PLGA@Cur NPs are shown in Appendix A. A distinctive peak at around 260 nm, seen in both spectra, was attributed to the presence of PLGA. A characteristic peak from Cur at 425 nm was also observed for the p-PLGA@Cur NPs. These results suggested successful encapsulation of Cur. The EEs of the PLGA@Cur and p-PLGA@Cur NPs were found to be 97.3 ± 1.2% and 93.8 ± 1.5%, and the drug loadings 8.9 ± 0.8% *w*/*w* and 8.6 ± 0.6% *w*/*w*, respectively.

The in vitro drug release profiles of the PLGA@Cur, p-PLGA@Cur and RBCM-p-PLGA@Cur NPs are given in Figure 2. The percentage of Cur released from the NPs after 48 h reached 80.8 ± 4.5% (≈ 139 μg) and 71.0 ± 4.3% (≈122 μg) for the p-PLGA NPs and RBCM-p-PLGA NPs, respectively, significantly higher than that from the monolithic PLGA NPs (28.3 ± 4.6%) (≈ 50 μg). The RBCM-p-PLGA NPs exhibited sustained release over 48 h. This prolonged release (cf. the p-PLGA@Cur NPs) can be attributed to the RBCM coating, which serves as a diffusional barrier to Cur release. 

The mechanism of drug release was explored using the Peppas equation [41]: *Q* = *kt^n^*. 

*Q* is the percentage of drug released at a given time, *t* is the release time, *k* is a rate constant, and *n* is an exponent that indicates the drug release mechanism. The outcomes of this analysis are summarized in Appendix A. The release exponents for p-PLGA@Cur and RBCM-p-PLGA@Cur were 0.606 and 0.595. Since these values are larger than 0.43 but smaller than 1.0, it appears that Cur was released through a combination of polymer erosion and drug diffusion. Based on these models, it would take around 980 h, 34 h, and 68 h for 100% release to be attained from the PLGA@Cur, p-PLGA@Cur and RBCM-p-PLGA@Cur NPs, respectively.

### 3.3. In Vitro Cytotoxicity 

The biocompatibility of the blank p-PLGA NPs and RBCM-p-PLGA NPs was evaluated by MTT analysis using L929 and H22 cells. As depicted in Figure 3a, both the p-PLGA and RBCM-p-PLGA NPs had minimal effect on the viability of the cells. The cell viability was more than 85%, even at p-PLGA and RBCM-p-PLGA NP concentrations of up to 500 μg/mL. The materials can thus be said to be highly biocompatible. Data on the in vitro cytotoxicity of the drug-loaded formulations on cancerous H22 cells are presented in Figure 3b. The half maximal inhibitory concentrations (IC_50_) of free Cur, p-PLGA@Cur NPs and RBCM-p-PLGA@Cur NPs to H22 cells were calculated to be 6.1 ± 0.5 μg/mL, 10.2 ± 0.8 μg/mL and 15.9 ± 1.3 μg/mL, respectively. When the Cur concentration was below 30 μg/mL, it appears that free Cur and the p-PLGA@Cur NPs were more toxic than the RBCM-p-PLGA@Cur NPs, which is likely a result of the sustained release of Cur caused by the RBCM shell. However, at Cur dosages above 50 μg/mL the RBCM-p-PLGA@Cur NPs exhibited more effective inhibition of H22 cell proliferation than p-PLGA@Cur NPs, which is thought to be because of enhanced endocytosis [42]. Collectively, these in vitro studies demonstrated that the RBCM-p-PLGA NPs are capable of efficiently inhibiting H22 cell proliferation.

### 3.4. Cellular Uptake In Vitro

CLSM analysis was used to evaluate the uptake of FITC labeled nanoparticles into H22 cells (Figure 4a). Strong FITC fluorescence appeared in the cells after incubation for 2 h with the RBCM-FITC-p-PLGA NPs. In sharp contrast, minimal FITC fluorescence was seen in the case of H22 cells treated with FITC-p-PLGA NPs. This is because both the RBCM coating on the NPs and the cancer cell membrane comprise phospholipid bilayers, enhancing cell endocytosis of the former over NPs lacking this coating. It appears that the RBCM coating was able to markedly enhance cellular uptake of the NPs. 4T1 cells were used to verify this hypothesis, and again much greater uptake was seen with the RBCM coated NPs (see Appendix A). The CLSM results obtained with RAW 264.7 cells showed weak FITC fluorescence (Figure 4b) after incubation with the RBCM coated NPs, however, and uptake here is much greater with the FITC-p-PLGA NPs. Cellular uptake is thus selective. The lack of uptake by RAW 264.7 (macrophage) cells arises because the biomimetic NPs express CD47 proteins from the RBCM at their exterior. These essentially mimic the surface properties of the host cell, so that they can evade uptake by macrophages and send a “don’t eat me” signal to the host [43]. Hence, the RBCM coating on the surface of the p-PLGA NPs can improve their biocompatibility. 

For H22 cells, the ability of the RBCM coating to enhance cellular uptake of the NPs is borne out by flow cytometry, which revealed that uptake of RBCM-FITC-p-PLGA NPs was around 1.6-fold greater than that of FITC-p-PLGA NPs (Figure 5). The inverse was seen with RAW 264.7 cells, where the uncoated NPs were taken up to a significantly greater extent than the RBCM-FITC-p-PLGA NPs (Appendix A). It is hence clear that the RBCM coating allows the NPs to be selectively taken up by cancerous cells and to evade phagocytosis by cells of immunity.

### 3.5. In Vivo Antitumor Efficacy

The antitumor efficiency of the NPs was investigated in H22-tumor bearing ICR mice. The change in tumor volume with time for mice given the different treatments is depicted in Figure 6a. The tumor volume for the RBCM-p-PLGA@Cur treatment group was lower than any other treatment group. Free Cur had relatively low efficacy, and at the end of the experimental period the tumor volume was only reduced by ca. 49.7 ± 3.2% compared to the negative control group. The p-PLGA@Cur NPs led to a tumor volume reduction of 72.3 ± 5.6%, while the RBCM-p-PLGA@Cur NPs led to an even greater reduction (87.6 ± 4.5%). The tumor size with the RBCM-p-PLGA@Cur NPs is significantly lower than with each of the other treatments (*p* < 0.005). It is thought that the RBCM coating on the surface of the p-PLGA NPs could enhance uptake by cancer cells and the accumulation of the formulation in the tumor. This is attributed to the erythrocyte membrane presenting the protein CD47, which allows the NPs to send a “don’t eat me” signal to the host, avoiding phagocytosis and persisting in the systemic circulation for an extended period of time [43].

Changes in body weight were also monitored (Figure 6b). Free Cur treatment led to notably decreased body weights, with the animal weights reduced by ca. 6.1 ± 1.2% at the end of the treatment period. This indicates off-site toxicity and side effects arose. The mice in the NP treatment groups experienced an increase in body weight with time, but this effect is smaller with the RBCM-p-PLGA@Cur treatment group. The RBCM coating appeared to permit targeting to the tumor, thus avoiding systemic toxicity from free Cur. Considering the saline group, the body weight of the mice increased significantly over the experimental period, presumably because of the growth of the tumor. After sacrifice at the end of the experiment, the tumors were excised and weighed (Figure 6c); the masses were in the order saline > free Cur > p-PLGA@Cur NPs > RBCM-p-PLGA@Cur NPs. Again, the mass of the tumors in the mice treated with the RBCM-coated NPs was significantly less (*p* < 0.05) than that seen with the other treatment groups. This is in accordance with the in vivo tumor volume data.

Histological analysis to assess tumor cell apoptosis was performed using an immunofluorescent TUNEL staining assay. The results are presented in Figure 7. In the animals that received pure Cur or the p-PLGA@Cur NPs, some green (apoptotic) cells were seen, but apoptosis was much more evident in the RBCM-p-PLGA@Cur NP treatment group. These in vivo results clearly show that the RBCM-p-PLGA@Cur NPs can act as an efficient drug delivery platform for tumor suppression.

To investigate the in vivo safety and biocompatibility of the RBCM-p-PLGA@Cur NPs, the major organs (heart, liver, spleen, lung, and kidneys) were recovered after sacrifice and imaged after H&E staining (Appendix A). No differences were observed between the negative control group and the animals receiving RBCM-p-PLGA@Cur NPs. The RBCM-p-PLGA@Cur NPs were hence determined to be non-toxic and safe for use in vivo. 

## 4. Discussion 

The results obtained here add to the growing body of evidence showing that using the RBCM as the exterior of NP delivery systems can increase the efficacy of anticancer therapeutics. Increased circulation times are observed with RBCM-coated NPs, including those made of PLGA [44], and this is expected to contribute to the in vivo efficacy of such systems. Qian et al. reported such an extended half-life and reduced tumor volumes for hydrogel-encapsulating paclitaxel-loaded red blood cell membrane nanoparticles [45], and Peng and co-workers noted analogous results with Prussian blue/manganese dioxide NPs coated with the RBCM [46]. Gao et al. used perfluorocarbon-loaded PLGA NPs coated with the RBCM to relieve hypoxia in the tumor and enhance radiotherapy, again observing extended circulation times and reduced tumor volumes [14]. Other authors have applied combined NP coatings containing both RBCM and cancer-cell membranes to doxorubicin-loaded CuS nanoparticles, with similar findings in vivo [47].

The RBCM-p-PLGA NPs platform reported here integrates two distinct materials: a polymer and a cell membrane, both of which play an important role in determining the overall properties of the composite. A previous report used the RBCM to coat docetaxel-loaded PLGA NPs, but found that the cumulative release of drug reached only around 20% after 72 h [48]. This is likely to be sub-optimal in terms of therapeutic efficacy, with much of the drug remaining in the formulation and thus not being therapeutically active. Our approach generated materials with a porous structure to overcome this limitation, and extended release systems were thereby obtained after the RBCM coating. 

The p-PLGA method proved to be an effective route to develop Cur-loaded NPs, and our materials are superior to other Cur delivery systems in the literature. For instance, Dwivedi et al. used a liquid-driven co-flow focusing process to fabricate Cur loaded PLGA NPs and obtained an encapsulation efficiency of approximately 70% [49]. In addition, Jamali et al. used a nanoprecipitation method to encapsulate Cur in PLGA NPs and the encapsulation efficiency and drug loading efficiency were calculated to be 27% and 4.5%, respectively [50]. Our systems are notably improved in terms of both encapsulation efficiency (>93.8%) and drug loading (>8%). Since all materials used to develop the new DDS reported in this work are clinically used or highly biocompatible, the RBCM-p-PLGA NP “artificial RBCs” should be able to act as safe and efficacious anticancer therapeutics.

## 5. Conclusions

In this study, curcumin (Cur)-loaded porous PLGA nanoparticles were developed. These were then cloaked with red blood cell membranes to prepare biomimetic platforms for targeted anticancer therapies. The resultant RBCM-p-PLGA@Cur NPs are spherical and around 160 nm in size, eminently suitable for uptake by cancer cells and retention in the tumor by means of the enhanced permeation and retention effect. They additionally exhibit sustained release over 48 h. While the drug-free particles were highly biocompatible (to both healthy and cancerous cells), the Cur loaded systems were effective in killing cancer cells in vitro, and also enjoyed enhanced tumor cell uptake over p-PLGA@Cur NPs. In contrast, the RBCM-coated NPs were not subject to phagocytosis by macrophage cells in vitro, while the blank p-PLGA@Cur NPs were vulnerable to this process. In H22-tumor bearing mice, the RBCM-p-PLGA@Cur NPs led to minimal systemic toxicity and clearly enhanced anti-cancer efficacy (in terms of tumor volume/mass and the presence of apoptotic cells) compared to either free Cur or p-PLGA@Cur NPs at an equivalent dose. The RBCM-p-PLGA@Cur NPs hence comprise a promising nanocarrier for the targeted treatment of cancer.

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
