# Peer review of "Erythrocyte Membrane Cloaked Curcumin-Loaded Nanoparticles for Enhanced Chemotherapy"

_pharmaceutics, 2019, doi:10.3390/pharmaceutics11090429_

Round 1
Reviewer 1 Report
The manuscript describes the fabrication of a biomimetic drug carrier, porous poly(lactic-co-glycolic acid) (PLGA) nanoparticles coated with red blood cell membranes, and its potential application in chemotherapy. I find the study comprehensive, with the extensive physicochemical and biological characterization of the material. However, I suggests some minor changes to be made before the manuscript can be accepted for publication:
1. The manuscript will benefit if the kinetics of the drug release is determined. The examples of kinetic models that can be used for the analysis of drug release from nanoparticles can be found here: Barzegar Jalali et al: Kinetic Analysis of Drug Release From Nanoparticles. J Pharm Pharmaceut Sci 11 (1): 167-177, 2008. When having the release model optimized, it would be possible to estimate the time needed for RBCM-p-PLGA NPs to reach 100% cumulative release.
2. I would also suggest presenting the data from Figure 2 in the form of the actual release, e.g. ug of drug released from the same mass of the carrier – just to compare if the therapeutic levels of drug are reached. This additional figure could be placed as the supplementary information.
3. Fig.3 lacks in the analysis of statistical significance of the data.
Author Response
Comments to author:
The manuscript describes the fabrication of a biomimetic drug carrier, porous poly (lactic-co-glycolic acid) (PLGA) nanoparticles coated with red blood cell membranes, and its potential application in chemotherapy. I find the study comprehensive, with the extensive physicochemical and biological characterization of the material.
We thank the reviewer for their kind words.
However, I suggest some minor changes to be made before the manuscript can be accepted for publication:
Point 1: The manuscript will benefit if the kinetics of the drug release is determined. The examples of kinetic models that can be used for the analysis of drug release from nanoparticles can be found here: Barzegar Jalali et al: Kinetic Analysis of Drug Release from Nanoparticles. J Pharm Pharmaceut Sci 11 (1): 167-177, 2008. When having the release model optimized, it would be possible to estimate the time needed for RBCM-p-PLGA NPs to reach 100% cumulative release.
Response 1: We thank the reviewer for this helpful comment and have added this analysis in the revised paper (see p8, Section 3.2, Table S1).
Point 2: I would also suggest presenting the data from Figure 2 in the form of the actual release, e.g. ug of drug released from the same mass of the carrier – just to compare if the therapeutic levels of drug are reached. This additional figure could be placed as the supplementary information.
Response 2: We thank the reviewer for this helpful suggestion. We have added some data to Section 3.2 (p8) of the revised paper.
Point 3: Fig.3 lacks in the analysis of statistical significance of the data.
Response 3: We thank the reviewer for this helpful suggestion. We have added statistical analysis to the revised paper (p9, Section 3.3).

Reviewer 2 Report
The authors stressed the significance of active targeting in the introduction section. However, there is no evidence that the described strategy belongs to the active targeting strategy. Please re-discuss this issue.
In figure4, RBCM-p-PLGA showed increased uptake rate comparing to p-PLGA in H22 cells. According to authors' discussion (in 3.4. section) this might be due to the interaction between RBCM and phospholipid bilayers of cancer cells. This explanation does not address the specific cancer targeting of the described strategy since phospholipid bilayers is not a cancer unique feature. Provide more convincing discussion.
Provide flow data for Figure4b.
Use more cell lines to verify the hypothesis.
Author Response
Comments to author:
Point 1: The authors stressed the significance of active targeting in the introduction section. However, there is no evidence that the described strategy belongs to the active targeting strategy. Please re-discuss this issue.
Response 1: Our work focuses more on biomimetic systems than on active targeting in the introduction section. Red blood cells have extended systemic circulation times, and being endogenous cells are highly biocompatible with low immunogenicity [1]. This is attributed to the erythrocyte membrane presenting the protein CD47, which can allow the NPs to avoid phagocytosis and persist in the systemic circulation for a long period of time [2] by sending a “don't eat me” signal to the host [3]. Therefore, it has a good targeting effect during the treatment process.
Point 2: In figure 4, RBCM-p-PLGA showed increased uptake rate comparing to p-PLGA in H22 cells. According to authors' discussion (in 3.4. section) this might be due to the interaction between RBCM and phospholipid bilayers of cancer cells. This explanation does not address the specific cancer targeting of the described strategy since phospholipid bilayers is not a cancer unique feature. Provide more convincing discussion.
Response 2. The reason for the NPs being taken up by cancer cells is because of the EPR effect, and uptake is greater for the RBCM-coated NPs vs the p-PLGA NPs owing to the former having a lipid bilayer membrane and the latter now. We have now additionally used 4T1 cells to demonstrate the ability of the RBCM coating to increase cellular uptake (please see p10 and Fig. S3).
Point 3: Provide flow data for Figure4b.
Response 3: Flow cytometry data has been included in Fig. S4.
Point 4: Use more cell lines to verify the hypothesis.
Response 4: We thank the reviewer for this helpful suggestion. We have now additionally used 4T1 cells to demonstrate that the ability of the RBCM coating can increase cellular uptake (please see p10 and Fig. S3).
References
Hayashi, K.; Yamada, S., et al. Red Blood Cell-Shaped Microparticles with a Red Blood Cell Membrane Demonstrate Prolonged Circulation Time in Blood. Acs. Biomater-Sci Eng. 2018, 4, 2729-2732, doi:10.1021/acsbiomaterials.8b00197. Willingham, S.B.; Volkmer, J.P., et al. The CD47-signal regulatory protein alpha (SIRPa) interaction is a therapeutic target for human solid tumors. P. Natl. Acad. Sci. USA. 2012, 109, 6662-6667, doi:10.1073/pnas.1121623109. Chambers, E.; Mitragotri, S. Long circulating nanoparticles via adhesion on red blood cells: Mechanism and extended circulation. Exp. Biol. Med. 2007, 232, 958-966

Reviewer 3 Report
The manuscript by Xie et al. describes assembly, loading with curcumin, and in vitro/in vivo applicaiton of biomimetic RBCM PLGa nanoparticles. The work is well designed and well written, and I would favorably see it published in Pharmaceutics provided that a couple of minor issues are addressed:
1) Row 66-67: actually, drug release from PLGA nanoparticles is a variegated phenomenon that strongly depends on different factors, such as PLGA composition and molecular weight. Release can occur with significantly higher kinetics, depending also on the nature of the payload. The author should better clarify and discuss this point.
2) row 329-330: Enhanced accumulation could for sure be responsible for this effect. Another possible explanation is that the different coating confers different capability to remain in the bloodstream. Ideally, a kinetic curve of nanoparticles coated and uncoated should be performed to evaluate this effect. Otherwise, the authors should at least discuss the possibility for different reasons fo this effect.
Author Response
Comments to author:
The manuscript by Xie et al. describes assembly, loading with curcumin, and in vitro/in vivo application of biomimetic RBCM PLGA nanoparticles. The work is well designed and well written, and I would favourable see it published in Pharmaceutics provided that a couple of minor issues are addressed:
We thank the reviewer for their kind words.
Point 1: Row 66-67: actually, drug release from PLGA nanoparticles is a variegated phenomenon that strongly depends on different factors, such as PLGA composition and molecular weight. Release can occur with significantly higher kinetics, depending also on the nature of the payload. The author should better clarify and discuss this point.
Response 1: We thank the reviewer for this helpful comment. The erythrocyte membrane-coated PLGA nanoparticles can extended systemic circulation times, and being endogenous cells are highly biocompatible with low immunogenicity [1]. When the erythrocyte membrane is destroyed by the lysosome, the porous structure of PLGA nanoparticles are exposed to sustained release the curcumin. The results are showed on (p8, Section 3.2)
Point 2: Row 329-330: Enhanced accumulation could for sure be responsible for this effect. Another possible explanation is that the different coating confers different capability to remain in the bloodstream. Ideally, a kinetic curve of nanoparticles coated and uncoated should be performed to evaluate this effect. Otherwise, the authors should at least discuss the possibility for different reasons for this effect.
Response 2: We thank the reviewer for this helpful comment. We have added some discussion on (p11, Section 3.5) of the revised paper.
References
Hayashi, K.; Yamada, S., et al. Red Blood Cell-Shaped Microparticles with a Red Blood Cell Membrane Demonstrate Prolonged Circulation Time in Blood. ACS. Biomater-Sci Eng. 2018, 4, 2729-2732, doi:10.1021/acsbiomaterials.8b00197.

Reviewer 4 Report
The manuscript deals with the preparation, characterizationa and evaluation of the in vitro/in vivo antitumoral activity of PLGA nanoparticles surface modified with red blood cell membranes.
The manuscript is well organised and the experimental section well conducted.
Only few suggestions and coment:
I suggest merging paragraph 2.2. "Characterization techniques" with paragraph 2.3 "Preparation and characterization of p-PLGA NPs.
In paragraph 2.4 "Loading of Cur" it is reported that only supernatant after centrifugation was measured to quantify the loading of Cur.
How was EE% and LC% calculated? According to the reported equation for EE% and LC% the mass of curcumin in nanoparticles should be detemined and not the concentration of the non-encapsulated Cur in the supernantant.
Author Response
Comments to author:
The manuscript deals with the preparation, characterization and evaluation of the in vitro/in vivo antitumoral activity of PLGA nanoparticles surface modified with red blood cell membranes. The manuscript is well organised and the experimental section well conducted. We thank the reviewer for their kind words.
Only few suggestions and comment:
Point 1: I suggest merging paragraph 2.2. "Characterization techniques" with paragraph 2.3 "Preparation and characterization of p-PLGA NPs.
Response 1: We thank the reviewer for this helpful comment, and have made these changes in the revised paper (see p4, Section 2.6).
Point 2: In paragraph 2.4 "Loading of Cur" it is reported that only supernatant after centrifugation was measured to quantify the loading of Cur.
How was EE% and LC% calculated? According to the reported equation for EE% and LC% the mass of curcumin in nanoparticles should be determined and not the concentration of the non-encapsulated Cur in the supernatant.
Response 2: We use the total Cur mass less the unloaded Cur present in the supernatant to determine the encapsulated Cur in the PLGA nanoparticles. We have corrected the formula in the revised paper (see p3, Section 2.3).

Round 2
Reviewer 2 Report
Authors addressed all reviewers' comments. I recommend the acceptance for the publication.